# FexSplice: A LightGBM-Based Model for Predicting the Splicing Effect of a Single Nucleotide Variant Affecting the First Nucleotide G of an Exon

**DOI:** 10.3390/genes14091765

**Published:** 2023-09-06

**Authors:** Atefeh Joudaki, Jun-ichi Takeda, Akio Masuda, Rikumo Ode, Koichi Fujiwara, Kinji Ohno

**Affiliations:** 1Division of Neurogenetics, Center for Neurological Diseases and Cancer, Nagoya University Graduate School of Medicine, 65 Tsurumai, Showa-ku, Nagoya 466-8550, Japan; atefeh.joudaki@gmail.com (A.J.); jtakeda@med.nagoya-u.ac.jp (J.-i.T.); amasuda@med.nagoya-u.ac.jp (A.M.); 2Department of Materials Science and Engineering, Nagoya University Graduate School of Engineering, Furo-cho, Chikusa-ku, Nagoya 464-8601, Japan; r1kum0.0d3@gmail.com (R.O.); fujiwara.koichi@hps.material.nagoya-u.ac.jp (K.F.)

**Keywords:** first nucleotide of an exon, splicing-affecting variants, LightGBM model, FexSplice web service program

## Abstract

Single nucleotide variants (SNVs) affecting the first nucleotide G of an exon (Fex-SNVs) identified in various diseases are mostly recognized as missense or nonsense variants. Their effect on pre-mRNA splicing has been seldom analyzed, and no curated database is available. We previously reported that Fex-SNVs affect splicing when the length of the polypyrimidine tract is short or degenerate. However, we cannot readily predict the splicing effects of Fex-SNVs. We here scrutinized the available literature and identified 106 splicing-affecting Fex-SNVs based on experimental evidence. We similarly identified 106 neutral Fex-SNVs in the dbSNP database with a global minor allele frequency (MAF) of more than 0.01 and less than 0.50. We extracted 115 features representing the strength of splicing *cis*-elements and developed machine-learning models with support vector machine, random forest, and gradient boosting to discriminate splicing-affecting and neutral Fex-SNVs. Gradient boosting-based LightGBM outperformed the other two models, and the length and nucleotide compositions of the polypyrimidine tract played critical roles in the discrimination. Recursive feature elimination showed that the LightGBM model using 15 features achieved the best performance with an accuracy of 0.80 ± 0.12 (mean and SD), a Matthews Correlation Coefficient (MCC) of 0.57 ± 0.15, an area under the curve of the receiver operating characteristics curve (AUROC) of 0.86 ± 0.08, and an area under the curve of the precision–recall curve (AUPRC) of 0.87 ± 0.09 using a 10-fold cross-validation. We developed a web service program, named FexSplice that accepts a genomic coordinate either on GRCh37/hg19 or GRCh38/hg38 and returns a predicted probability of aberrant splicing of A, C, and T variants.

## 1. Introduction

Pre-mRNA splicing is a fundamental process in eukaryotic gene expression that involves the precise removal of introns and the joining of exons to generate mature mRNA. Splicing is mediated by the spliceosome, a dynamic and highly regulated macromolecular complex consisting of target pre-mRNA, small nuclear ribonucleoproteins (snRNPs), and numerous other proteins. The spliceosome catalyzes splicing in two steps. In the first step, the spliceosome is assembled on pre-mRNA, where the intron/exon and exon/intron boundaries comprised specific *cis*-elements including the 5′ splice site (ss), 3′ ss, polypyrimidine tract (PPT), and branch point sequence (BPS) are recognized by *trans*-acting RNA-binding proteins (RBPs) such as the snRNPs, heterogeneous nuclear ribonucleoproteins (hnRNPs), and serine arginine-rich splicing factors (SRSFs) [1,2]. Single nucleotide variations (SNVs) that disrupt *cis*-acting splicing elements and compromise catalytic functions of *trans*-acting RBPs impair finely tuned alternative and constitutive splicing events [3]. Disruptions in splicing have been implicated in a wide range of diseases including cancer, neurodegenerative disorders, and Mendelian disorders such as congenital myasthenic syndromes [4].

The BPS and PPT are first recognized by SF1 and U2AF65, respectively [5]. Introns with a long PPT do not require the binding of U2AF35 to the intron–exon boundary because U2AF65 is able to bind to PPT strongly, which is called an AG-independent 3′ ss (Figure 1). Conversely, introns with a short or degenerate PPT require the binding of U2AF35 to the intron–exon boundary to reinforce the binding of U2AF65 to PPT, which is called an AG-dependent 3′ ss. We previously reported that SNVs affecting the first nucleotide G of an exon (Fex-SNVs) cause aberrant splicing at the AG-dependent 3′ ss’s but not at the AG-independent 3′ ss’s [6], which has also been proven at the structural level by others [7]. Serial mutagenesis to gradually increase the length of PPT revealed that a stretch of pyrimidines in PPT needs to be 10 to 15 nucleotides or more to make the 3′ ss insensitive to a Fex-SNV [6]. When the first nucleotide of an exon is not G in the reference sequence, binding of U2AF35 to the intron–exon boundary is predicted to be weak, and such 3′ ss’s are mostly AG-independent [8].

Although the AG-dependence of the 3′ ss predicts the splicing effects of Fex-SNVs, there is no dependable rule to determine the AG-dependence of the 3′ ss. Several prediction tools such as SpliceAI [9] and Collapsed Isoform SpliceAI (CI-SpliceAI) [10] have been developed to predict the splicing consequences of SNVs. However, these tools were not optimized for predicting the splicing effects of Fex-SNVs. We previously developed web service programs of support vector machine (SVM)-based IntSplice (https://www.med.nagoya-u.ac.jp/neurogenetics/IntSplice_v1.0/) (accessed on 1 August 2023) [11] and gradient boosting-based IntSplice2 (https://www.med.nagoya-u.ac.jp/neurogenetics/IntSplice2/) (accessed on 1 August 2023) [12], both of which predict the splicing effects of intronic SNVs at positions −50 to −3, but do not cover Fex-SNVs. To address this challenge, we first curated a dependable dataset that comprised Fex-SNVs and their splicing effects by scrutinizing available articles, and developed a machine-learning model, FexSplice, using Light Gradient Boosting Machine (LightGBM) [13] dedicated to predicting the splicing effects of Fex-SNVs. We hope that FexSplice sheds light on frequently underestimated splicing-affecting Fex-SNVs.

## 2. Materials and Methods

### 2.1. Fex-SNV Dataset

We scrutinized Fex-SNVs in the Human Gene Mutation Database (HGMD) Professional released in April 2020 [14], the ClinVar released on 15 March 2021 [15], and PubMed including a recently published article on splicing variants [16]. We only collected Fex-SNVs with G as the first nucleotide of an exon in the reference sequence.

For HGMD Pro, we chose disease-associated SNVs in the mutation categories of DM (disease-causing mutation) and SM (splicing mutation). For ClinVar 2021 [15], we chose disease-associated SNVs with CLNSIG = pathogenic. We thus identified 801 Fex-SNVs according to the transcript annotations of Ensembl release 101 [17]. We first eliminated Fex-SNVs in the first and last exons because these exons had no upstream and downstream sequences, respectively, and some features could not be extracted from these exons. The predicted amino acid substitutions of Fex-SNVs were annotated in HGMD Pro, ClinVar, and the literature, but their effects on pre-mRNA splicing, if any, remained mostly unannotated except for the literature. We thus scrutinized the experimental details of available articles to accurately annotate Fex-SNVs. A Fex-SNV was recognized as splicing-affecting when aberrant splicing was demonstrated using RT-PCR of either the patient sample or a minigene construct. If RefSeq [18] shows two or more splicing isoforms at a Fex-SNV, the Fex-SNV was included when authors addressed which splicing isoform was affected by the Fex-SNV. In contrast, when authors did not address the splicing isoforms, the Fex-SNV was excluded from our dataset. These filtrations reduced the number of Fex-SNVs to 106 splicing-affecting and 5 neutral Fex-SNVs in HGMD Pro, ClinVar, and the literature (Appendix A).

For additional neutral Fex-SNVs, we extracted 1005 Fex-SNVs from dbSNP (build 151) on GRCh37/hg19 [19]. The 1005 neutral Fex-SNVs were first filtered by a global minor allelic frequency (MAF) greater than 0.01 and less than 0.5, which produced 156 neutral Fex-SNVs. MAF > 0.5 indicates that the reference nucleotide is minor. To match the numbers of splicing-affecting and neutral Fex-SNVs, we randomly selected 101 out of 156 neutral Fex-SNVs. In the selection, we attempted to exclude Fex-SNVs with similar flanking sequences or neutral Fex-SNVs identified in the course of disease analysis. By adding 5 neutral Fex-SNVs in HGMD Pro, ClinVar, and the literature stated above, we obtained 106 neutral Fex-SNVs (Appendix A).

### 2.2. Extraction of Features

We first extracted 115 features dictating the strength of splicing *cis*-elements, most of which were used to predict the splicing effects of intronic SNVs (IntSplice [11] and IntSplice2 [12]) (Appendix A). The 115 features included the followings. First, the best BPS was searched for between Int^−50^ to Int^−3^ using the yUnAy motif [20]. The position weight matrix score as well as the conserved branch point “A” nucleotide were evaluated. Second, the length of PPT as well as the ratios of T, G, purines (A/G), and pyrimidines (C/T) in PPT were evaluated. As GGG trinucleotides are frequently recognized by splicing-suppressing hnRNP H and hnRNP K [21,22], the presence of GGG in PPT was evaluated. Third, we previously observed that nucleotides at Int^−7^, Int^−6^, Int^−5^, and Int^−3^, as well as Ex^+2^ and Ex^+3^, play critical roles in splicing [11]. We included these nucleotides in our features. Fourth, SD-Score at the 5′ ss [23], MaxEntScan scores at the 3′ and 5′ ss’s [24], and Shapiro Senapathy scores [25] at the 3′ and 5′ ss’s were included as integrated measures to evaluate the strength of constitutive splicing *cis*-elements. Fifth, RBPs exert essential roles in both alternative and constitutive splicing events [26,27]. In our previous machine-learning model, IntSplice [12], to predict the splicing effects of intronic SNVs, we showed that the inclusion of RBP-biding sites markedly improved the performance. We thus included the sum scores of SpliceAid2 [28] of 71 RBPs in our features. As we could not predict which specific feature best dictated the strength of splicing signals, we admitted multicollinearity of features. Spearman’s rank correlation coefficients of all available pairs of 115 features are indicated in Appendix A.

### 2.3. Machine-Learning Models

We generated machine-learning models with SVM (LinearSVC) [29], random forest (RandomForest) [30], and gradient boosting (LightGBM) [13]. For each model, we optimized hyperparameters using grid search. Feature importance was obtained from each modeling tools with default settings. We also eliminated features one by one using a method of meta-transformer for selecting features based on importance weights [31] by leave-one-out cross-validation (LOOCV). The performance of each model was evaluated by the area under the receiver operating characteristic curve (AUROC), the area under the precision recall curve (AUPRC), and seven statistical measures recommended by the Human Mutation Guidelines (see a legend of Table 1 for details) [32,33]. As we included all the identified splicing-affecting Fex-SNVs in our dataset, we did not create a separate test dataset. Instead, we employed leave-one-out or 10-fold cross-validation.

## 3. Results

### 3.1. Generation of Models with LinearSVC, Random Forest, and LightGBM

In this study, we generated machine-learning models to predict whether a Fex-SNV affecting the G nucleotide at the first nucleotide of an exon affects splicing or not. We first created a curated dataset of Fex-SNVs that comprised 106 splicing-affecting and 106 neutral Fex-SNVs (Appendix A). For each Fex-SNV, we extracted 115 features that dictated the strength of splicing *cis*-elements (Appendix A). We then generated three machine-learning models: LinearSVC [29], RandomForest [30], and LightGBM [13]. Each model was evaluated by AUROC and AUPRC (Figure 2), as well as seven statistical measures (accuracy, precision, recall/sensitivity, specificity, F1 score, NPV, and MCC) using 10-fold cross-validation (Table 1). LightGBM produced the highest AUROC and the highest scores in six out of the seven statistical measures except for specificity. The importance of 115 features by LightGBM were inspected using 10-fold cross-validation (Figure 3) and will be discussed in detail in the Discussion section.

**Table 1 genes-14-01765-t001:** Comparison of nine statistical measures using 10-fold cross-validation of LinearSVC, RandomForest, and LightGBM models with 115 features, as well as a LightGBM model with 15 features.

Model	LinearSVC (115)	Random Forest (115)	LightGBM (115)	LightGBM (15)
**Accuracy ^1^**	0.64 ± 0.10	0.71 ± 0.07	0.75 ± 0.09	0.77 ± 0.07
**Precision ^2^**	0.64 ± 0.09	0.71 ± 0.07	0.77 ± 0.11	0.80 ± 0.12
**Recall ^3^**	0.65 ± 0.15	0.73 ± 0.12	0.74 ± 0.14	0.78 ± 0.13
**Specificity ^4^**	0.63 ± 0.11	0.70 ± 0.08	0.78 ± 0.13	0.77 ± 0.15
**F1 score ^5^**	0.64 ± 0.11	0.71 ± 0.08	0.75 ± 0.10	0.77 ± 0.07
**NPV ^6^**	0.65 ± 0.11	0.73 ± 0.11	0.76 ± 0.11	0.79 ± 0.11
**MCC ^7^**	0.29 ± 0.19	0.43 ± 0.15	0.52 ± 0.18	0.57 ± 0.15
**AUROC**	0.69 ± 0.08	0.79 ± 0.08	0.84 ± 0.08	0.86 ± 0.08
**AUPRC**	0.71 ± 0.08	0.82 ± 0.07	0.85 ± 0.08	0.87 ± 0.09

The number of features is indicated in parentheses. Mean and SD are indicated. ^1^ Accuracy, overall correctness of the classifier: Accuracy = (TP + TN)/(TP + TN + FP + FN); ^2^ Precision (positive predictive value), correctness of positive predictions: Precision = TP/(TP + FP); ^3^ Recall (sensitivity or true positive rate), classifier’s ability to identify positive instances: Recall = TP/(TP + FN); ^4^ Specificity (true negative rate), classifier’s ability to identify negative instances: Specificity = TN/(TN + FP); ^5^ F1 Score, balanced metric considering false positives and negatives: F1 Score = 2 * (Precision * Recall)/(Precision + Recall); ^6^ NPV (negative predictive value), correctness of negative predictions: NPV = TN/(TN + FN); ^7^ MCC (Matthews correlation coefficient), balanced measure considering all values in the confusion matrix: MCC = (TP * TN − FP * FN)/((TP + FP) * (TP + FN) * (TN + FP) * (TN + FN))1/2. TP (true positive) and FN (false negative) are the numbers of splicing-affecting Fex-SNVs that were predicted to be splicing-affecting and neutral, respectively. FP (false positive) and TN (true negative) are the numbers of neutral Fex-SNVs that were predicted to be splicing-affecting and neutral, respectively.

We next eliminated features one-by-one from the three models using LOOCV (Appendix A). Neither LinearSVC nor RandomForest reasonably improved the balanced accuracy by eliminating features. In contrast, the balanced accuracy was maximized at 15 features with LightGBM. Elimination of features from 115 to 15 increased the AUROC of LightGBM model from 0.84 ± 0.08 (mean and SD) to 0.86 ± 0.08 (Figure 2E,G and Table 1). Similarly, elimination of features increased in all the seven statistical measures of LightGBM model by approximately 2% (Table 1).

As expected, the feature importance values of the 15-feature-based LightGBM model using 10-fold cross-validation (Appendix A) were similar to those of the 115-feature-based model using 10-fold cross-validation (Figure 3). We herein refer to the 15-feature-based LightGBM model as FexSplice.

### 3.2. Comparison of FexSplice with SpliceAI and CI-SpliceAI

SpliceAI [9] predicts the positions of ss’s using the residual neural networks (ResNet) trained with a 10 Kbp segment annotated in the GTEx database. CI-SpliceAI [10] is based on the SpliceAI and retrained using a collapsed isoform set representative of all manually annotated constitutive and alternative splice sites in GENCODE. SpliceAI [9] and CI-SpliceAI [10] are also able to predict the splicing effects of Fex-SNVs. We calculated the AUROC, the AUPRC, and seven statistical measures of SpliceAI and CI-SpliceAI with our dataset (Appendix A). FexSplice was trained with our dataset, whereas SpliceAI and CI-SpliceAI were not. Thus, statistical measures of SpliceAI and CI-SpliceAI cannot be unbiasedly compared with those of FexSplice. Nevertheless, precision and specificity were better in SpliceAI and CI-SpliceAI compared to those in FexSplice. This was at the cost of a much lower recall value of 0.22 in both SpliceAI and CI-SpliceAI compared to 0.78 ± 0.13 (mean and SD) in FexSplice. As SpliceAI and CI-SpliceAI were developed to identify ss’s in a large number of candidates in the whole genome, they were likely to be designed to reduce false positives. This may account for high precision and specificity values with low recall values in SpliceAI and CI-SpliceAI.

### 3.3. Web Service of FexSplice

We developed a web service program, FexSplice, (https://www.med.nagoya-u.ac.jp/neurogenetics/FexSplice) (accessed on 1 August 2023) (Figure 4). The FexSplice web service accepts a genomic coordinate in either GRCh37/hg19 or GRCh38/hg38 and maps it to all the annotated coding transcripts in Ensembl release 101. FexSplice analyzes all the transcripts and generates three possible Fex-SNVs at the given coordinate. LightGBM automatically generates a probability score for each Fex-SNV with 0.5 being the threshold. The default threshold of 0.5 by LightGBM was used in FexSplice. Fex-SNVs with a probability less than 0.5 are predicted to be splicing-insensitive, while those with a probability of 0.5 or more are predicted to be splicing-affecting. When two or more transcripts exist at Fex-SNV, FexSplice predicts the effects of splicing for all the relevant transcripts. Pre-processed genome-wide FexSplice dataset was generated on GRCh37/hg19, and was converted to the GRCh38/hg38 version using LiftOver [34], both of which are downloadable from the FexSplice web site.

## 4. Discussion

Our study aimed to develop a model to predict the splicing effect of Fex-SNVs. We scrutinized available articles and curated a dataset that comprised 106 splicing-affecting and 106 neutral Fex-SNVs (Appendix A). For each Fex-SNV, 115 features dictating the strength of splicing signals were extracted (Appendix A). Evaluation of the discrimination models by LinearSVC, RandomForest, and LightGBM using 10-fold cross-validation showed that LightGBM produced the highest AUROC, the highest AUPRC, and the highest scores in six out of the seven statistical measures (Table 1). Elimination of the least important feature one-by-one using cross-validation showed that the performance of LightGBM models became the best with 15 features (Appendix A).

We evaluated the importance of 115 features (Figure 3) and 15 features (Appendix A) both using 10-fold cross-validation and found that highly ranked features were similar between the two models. As our features had multicollinearity (Appendix A), high feature importance did not exclusively represent essential features. Nevertheless, the following features were critical. First, among the 115 features (Figure 3), the ratio of T nucleotides in PPT was ranked first and its importance was markedly higher than the other features. The preference of T over C in PPT was previously reported [35,36]. Similarly, the ratio of G nucleotides in PPT was ranked fifth. A more deleterious effect of G than A in PPT on binding to U2AF65 was also previously reported [37]. Additionally, three other features for PPT and four features for BPS are included in the top 30 features. The importance of PPT in the discrimination models is in accordance with the notion that the AG-dependent 3′ ss’s are vulnerable to Fex-SNV. Second, MaxEntScan::5′ss [24], SD-score [23], and Shapiro Senapathy score at 5′ ss [25], all of which represented the splicing signals at the 5′ ss, were ranked second, eleventh, and twelfth, respectively. Unexpectedly, MaxEntScan::5′ss had a higher importance than MaxEntScan::3′ss, which was ranked seventh. The importance of the splicing signals at the 5′ ss is likely to support the exon-recognition model, in which an exon not an intron is recognized as a single unit in pre-mRNA splicing [38]. Third, eight of the top 30 features were for the presence of RBP-binding sites. RBPs exert essential roles in both alternative and constitutive splicing events [26,27]. As indicated in Section 2.2, we previously showed that the inclusion of RBP-biding sites markedly improved the performance of IntSplice, a tool to predict the splicing effects of intronic SNVs [12]. Among the eight RBPs, ETR-3 (CELF2) and MBNL1 were ranked eighth and tenth, respectively. Abnormal downregulation of MBNL and upregulation ETR-3 are hallmarks of myotonic dystrophy, and their effects on pre-mRNA splicing have been extensively studied [39]. However, myotonic dystrophy was not included in either the title or the abstract of any article showing splicing-affecting Fex-SNVs (Appendix A). In addition, ETR-3-binding sequences according to SpliceAid2 were observed in 18 out of 106 splicing-affecting and 21 out of 106 neutral Fex-SNVs (*p*-value = 0.72 by Fisher’s exact test). Similarly, MBNL1-binding sequences were observed in 21 out of 106 splicing-affecting and 27 out of 106 neutral Fex-SNVs (*p*-value = 0.41). Thus, the high feature importance values of ETR-3 and MBNL1 were unlikely to be accounted for by reporting bias of splicing-affecting Fex-SNVs. Although the binding of hnRNP A1 was not included in the top 30 features, hnRNP A1 directly binds to the 3′ ss of *SMN2* exon 7 and suppresses its splicing [40]. However, RBPs are unlikely to bind to the 3′ ss where core spliceosomal components assemble. Thus, the presence of binding sites for RBPs is likely to represent that the splicing signals on and around the exon are weak and that the binding of RBP(s) is required for the exon recognition. Fourth, exonic features such as the exon length and the first-to-third exonic nucleotides played essential roles. We unexpectedly observed that out of the 12 exonic and 12 intronic nucleotides in the 115 features (Appendix A), four exonic nucleotides (T at Ex^+1^, A at Ex^+1^, C at Ex^+3^, and G at Ex^+2^) were included in the top 30 features, whereas only one intronic nucleotide (T at Int^−5^) was included. Aberrant splicing due to T at Ex^+1^ rather than A at Ex^+1^ was previously reported [41]. Crystal structure of U2AF1 (U2AF35) bound to the 3′ ss showed that a nucleotide at Ex^+2^ was not strictly recognized by U2AF1 and a nucleotide at Ex^+3^ was not bound by U2AF1 [7]. Nevertheless, C at Ex^+3^ and G at Ex^+2^ were included in the top 30 features. We previously showed that G at Int^−3^ was markedly detrimental for pre-mRNA splicing, and A at Int^−3^ followed [11]. However, neither nucleotide was included in the top 30 features, which was likely to be masked by multicollinearity of 115 features.

Comparison of FexSplice with SpliceAI and CI-SpliceAI showed that FexSplice outperformed the others in seven out of the nine statistical measures, although FexSplice should be biased by overfitting to our dataset compared to the others. To fairly compare the performance of different tools, models should be generated by an identical training dataset and evaluated by an identical testing dataset, as we previously performed for InMeRF, a tool for predicting the pathogenicity of missense SNVs [42]. We, however, did not recapitulate the generation of models with SpliceAI and CI-SpliceAI. We suppose that the splicing effects of Fex-SNVs have been underestimated in identifying pathogenic variants in human diseases. We hope that FexSplice will help disclose yet unidentified splicing effects of Fex-SNVs, and also understand the physiological mechanisms of the recognition of the 3′ ss’s.

## Figures and Tables

**Figure 1 genes-14-01765-f001:**
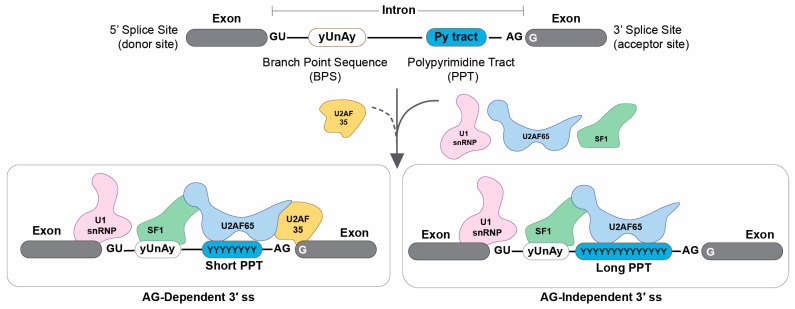
AG-dependent and AG-independent 3′ splice sites (ss’s). Introns with a short or degenerate PPT require both U2AF65 and U2AF35 for the recognition of the 3′ ss, which is called the AG-dependent 3′ ss. Introns with a long stretch of PPT strongly bind to U2AF65 and do not require binding of U2AF35, which is called the AG-independent 3′ ss. The 3′ ss’s without a G at the first nucleotide of an exon in the reference sequence are mostly AG-independent.

**Figure 2 genes-14-01765-f002:**
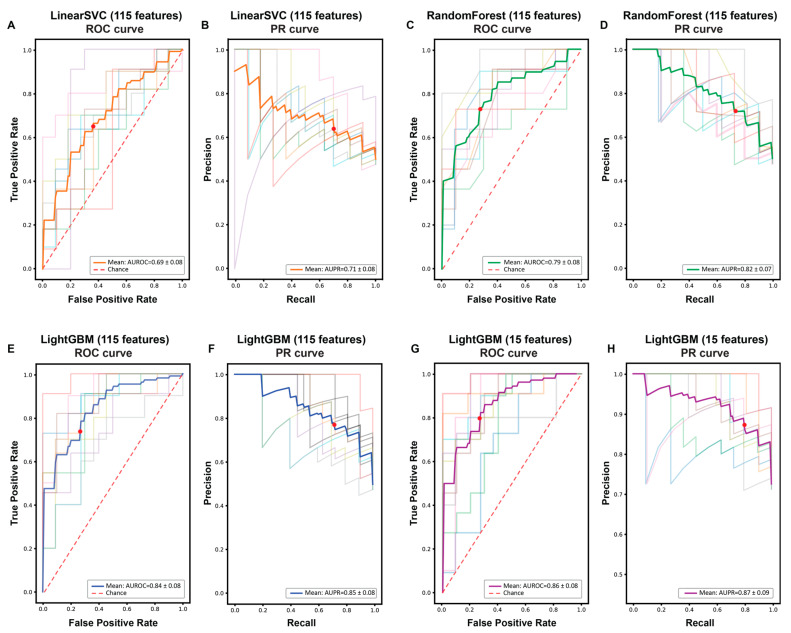
Receiver operating characteristics (ROC) (**A**,**C**,**E**,**G**) and precision recall (PR) (**B**,**D**,**F**,**H**) curves of LinearSVC (**A**,**B**), RandomForest (**C**,**D**), and LightGBM (**E**,**F**) models with 115 features, as well as a LightGBM (**G**,**H**) model with 15 features, all using 10-fold cross-validation. Thin lines represent each of the 10-fold validations, and thick lines represent 10-fold cross-validation. Red dots indicate where the threshold of pathogenic probability is set to 0.5.

**Figure 3 genes-14-01765-f003:**
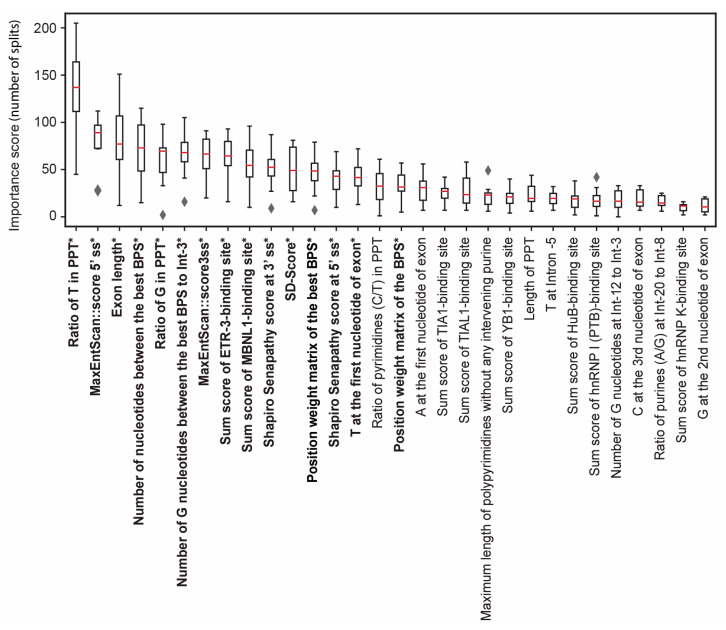
The top 30 features, ranked by their importance, are displayed along with associated median and interquartile range values. This ranking is derived from the feature importance analysis using 10-fold cross-validation of a LightGBM model trained with 115 features. Bold letters with an asterisk indicate 15 features that maximized the AUROC in recursive feature elimination (Appendix A), which were used to generate FexSplice.

**Figure 4 genes-14-01765-f004:**
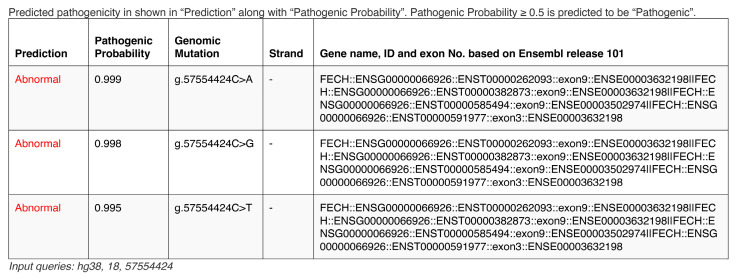
An example output of the FexSplice web service (https://www.med.nagoya-u.ac.jp/neurogenetics/FexSplice, accessed on 1 August 2023). G.57554424C>A on chromosome 18 (GRCh38/hg38) in *FECH* was previously reported to cause aberrant splicing [6]. The chromosome number and genomic coordinate were entered into the FexSplice web service. Predicted pathogenicity (abnormal in red letters and normal in black letters) and its probability were returned for three possible Fex-SNVs. Pre-processed genome-wide FexSplice datasets on GRCh37/hg19 and GRCh38/hg38 are also available. For g.57554424C>A, SpliceAI predicted a moderate effect on acceptor loss (Δ score = 0.45) and CI-SpliceAI predicted a minor effect on acceptor loss (Δ score = 0.24).

## Data Availability

The data presented in this study are available upon request to the corresponding author.

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
