# Peer review of "FexSplice: A LightGBM-Based Model for Predicting the Splicing Effect of a Single Nucleotide Variant Affecting the First Nucleotide G of an Exon"

_genes, 2023, doi:10.3390/genes14091765_

Round 1
Reviewer 1 Report
In their manuscript titled “FexSplice: a LightGBM-based model for predicting the splicing effect of a single nucleotide variant affecting the first nucleotide G of an exon”, Joudaki et al. present a new tool for evaluating potential splicing disruption in the human genome at the first G nucleotide of exons. The authors test LinearSVC, Random Forest and LightGBM models, and refine the LightGBM model, which is the best performing, to the 15 most-important features. They developed a web interface that allows public access to the FexSplice tool. They provide an example of g.57554424 on Ch.18 in hg38 for how FexSplice can predict abnormal splicing caused by Fex-SNVs. This manuscript addresses an important challenge, namely predicting SNVs that can result in disease-relevant alternative splicing defects. The specific case their model addresses is an important subset of SNVs. Further, the results of narrowing important features from 115 down to 15 can offer some potential insight into the splicing process. The manuscript is well written and easy to follow. The manuscript will be of interest to bioinformaticians, scientists working with potential human splice variants, and possibly scientists studying the mechanism of exon recognition. My comments to improve the manuscript are below.
1. Line 179: The figure reference should be Figure 2E, G and Table 1, rather than Figure 2A.
2. The Supplemental Tables are not included in the submission, and thus it is not possible to evaluate this data. All of the missing tables and figures are listed in Lines 283-288. In particular Figure S1 and Table S3 are missing, and are discussed both in the results and in the discussion (Lines 274-276).
3. Line 206 – there is a dash in the middle of the URL such that it doesn’t work properly. The website needs to have the citation of the paper or pre-print (once accepted). The website should also include a more detailed description of the derivation of the “pathogenic probability” and what this statistic means, as well as the datasets used to train the algorithm. The button on the results page should be “Rerun” instead of “Retrun”. On the positive site, the website works and is publicly accessible.
4. How was the probability threshold of 0.5 in Line 212 calculated? Is there empirical data that can be used to justify this threshold? And by splice-affecting, does this correlate with a particular Psi value?
5. Figure 3 – It is very interesting that the sum score of MBNL1 and ETR-3 binding sites has such a significant effect on improving the performance of the algorithm. The authors suggest this demonstrates that the binding of RBPs is required for exon recognition, as splicing signals are weak. I do not disagree and find this data intriguing, yet the sites that are most important are AS factors, rather than the binding of more constitutive RBPs (like their cited example of hnRNP A1). CELF and MBNL proteins are known alternative splicing regulators, so does this result suggest they also have a dedicated role more generally in regulating the splicing of exons with weak splicing signals? Can this really be generalized genome wide? Or does this rather reflect biases in the Human Gene Mutation Database (HGMD) and ClinVar, and thus the training dataset, based on the number of patients with muscle-related diseases that may be represented in those databases? Also, the training set was selected based on verified missplicing events, and perhaps this dependence on MBNL and ETR-3/CELF2 is reflected in a broader literature analyzing splicing defects in patients with neuromuscular disease, and underrepresentation of other disease categories.
6. The manuscript is missing a clear example of real data and use of FexSplice to identify novel splicing mutations. Was a test dataset employed? Figure 4 shows an example, but it isn’t clear if this example is not predicted to be a splicing mutation in SpliceAI or CI-SpliceAI, and there isn’t a representative figure proving that this even actually leads to missplicing. Is there evidence for this event in other patient data available in public databases? The context of the significance of this finding is missing.
7. How does FexSplice perform on the training dataset? Are all 106 splice-affecting SNVs recognized? How many of the 101 neutral SNVs are also called?
8. The manuscript is missing more detail on the comparison of FexSplice with two other tools. This should be in Supplementary Table S3, which is not included. What is the advantage of FexSplice over these other tools? Is there any way to estimate how many false positives FexSplice returns? FexSplice has a much higher recall but lower precision and specificity than SpliceAI and CI-SpliceAI, so what does this mean in a practical sense, or how does this limit the utility of FexSplice? This is important information to include for biomedical scientists who might want to use this software (as the authors clearly intend).
9. Line 279-280: Does this sentence mean the authors tried to recapitulate SpliceAI or CI-SpliceAI and were unable to, or that this is beyond the scope of this work? This would be an important difference in the meaning of this sentence.
There are a few minor grammar errors, but the manuscript is well-organized, clearly written and easy to understand.
Reviewer 2 Report
Review on the manuscript titled “FexSplice: a LightGBM-based model for predicting the splicing 2
effect of a single nucleotide variant affecting the first nucleotide G of an exon”.
The manuscript addresses the preferential first exon base being ‘G’ in AG dependent U2AF35 binding, and is a follow-up of the authors’ original article published in 2011 (PMID: 21288883). Authors selected AG dependent exons with short Polypyrimidine tracks and checked them against the mutation databases for their SNV status. Consequently, they provided online tool Fexsplice for calculating exon skipping chance due to the Fex polymorphism with high specificity and selectivity by applying 3 deep learning tools: LinearSVC, RandomForest, and LightGBM against 150 features on the 2 samples (AG dependent/independent exons), then scoring and reducing the number of features down to 15. Lastly, they chose LightGBM as a target tool for prediction due to the best Roc/PM curves quality score on the squeezed set of features (15), yielding the Roc of 0.86.
Overall, the problem statement, methods, data, and paper’s logical structure and quality are quite acceptable. The results would be relevant for those in the field. The comments are listed below.
1) grading boosting (p.3 s127) -> gradient boosting;
2) “Figure 3. Feature importance of a LightGBM model with 115 features by 10-fold cross validation. Feature importance of the top 30 features is plotted with the median and the interquartile range. Bold letters with an asterisk indicate 15 features that maximized the AUROC in feature elimination” – it looks like the first sentence in Fig. title is a bit misplaced. Better start with the second statement with first statement put on the second (third) place. Also: “Feature importance” -> “Feature importance ranking from the top”. Also, it may be relevant to note that “Bold letters with an asterisk indicate 15 features that maximized the AUROC in feature elimination” in LightGBM tool.
The language renders minor revisions.
Round 2
Reviewer 1 Report
The authors have addressed all of my comments. The text edits and figure edits have greatly improved the clarity of the manuscript, as well as the usability and clarity of the website.